# Maximal Respiratory Muscle Strength in Singaporean Adults: Normative Reference Values and Predictive Models from a Cross-Sectional Study

**DOI:** 10.3390/muscles4040047

**Published:** 2025-10-20

**Authors:** Katherin S. Huang, Filzah Diana Binte Roslan, Hong Hin Cheong, Jeremy J. Woo, Kian Wee Tan, Kim Swee, Shu-Ying Liang, Wei Xi Lau, Meredith T. Yeung

**Affiliations:** 1Professional Officers Division, Singapore Institute of Technology, 11 New Punggol Road, Singapore 828616, Singapore; 2Health and Social Sciences Cluster, Singapore Institute of Technology, 11 New Punggol Road, Singapore 828616, Singapore

**Keywords:** maximal inspiratory pressure (MIP), maximal expiratory pressure (MEP), respiratory muscle strength, reference values, regression equations

## Abstract

Maximal inspiratory pressure (MIP) and maximal expiratory pressure (MEP) are commonly used proxies to measure respiratory muscle strength. Current literature lacks recent large-scale normative reference values (NRV) for these pressures in healthy Singaporean adults. Moreover, no consensus exists on the variables that could influence MIP and MEP. This cross-sectional study aims to: (1) establish the NRV of MIP and MEP; (2) determine the correlations of variables that could influence these pressures; and (3) develop regression equations using non-spirometry variables to estimate reference values for MIP and MEP in the healthy Singaporean population aged 21 to 80 years. MIP and MEP were measured alongside demographic and anthropometric data collected from 391 participants (202 females, 189 males) recruited via convenience sampling. Median MIP and MEP values were significantly higher in males (112 and 85 cmH_2_O) than in females (83 and 64 cmH_2_O). Spearman correlations revealed significant associations between MIP/MEP and gender, height, weight, body mass index (BMI), waist–hip ratio, and spirometric variables. Regression models using age, gender, height, and weight explained 27.5% and 32.1% of the variance in MIP and MEP, respectively. This study updated the NRV of MIP and MEP and provided practical predictive equations for assessing respiratory muscle strength in Singapore.

## 1. Introduction

Maximal inspiratory pressure (MIP) and maximal expiratory pressure (MEP) are commonly used proxies for measuring respiratory muscle strength. The measurement technique is standardised in international guidelines, such as the American Thoracic Society (ATS) and the European Respiratory Society (ERS) [1], which recommend using a pressure manometer or a spirometer with dedicated MIP and MEP functions, along with consistent patient posture, instruction and effort to ensure accuracy and reproducibility. These measures are essential in the clinical evaluation of neuromuscular disorders, chronic respiratory diseases, and in monitoring the effects of therapeutic intervention, such as inspiratory muscle training. While MIP and MEP are clinically useful measures of respiratory muscle strength, their interpretation requires population-specific normative reference values (NRV) due to the influence of measurement techniques, as well as demographic, anthropometric and health or lifestyle factors. NRV are statistically derived ranges that define expected measurements within a healthy population. These values provide a basis for comparing individual results to determine normality or possible impairment. Even with international standardised measurement protocols, normative values can vary across populations [1,2].

Developing population-specific NRV helps to account for these inherent differences, reducing interpretative errors and control variability unrelated to the measurement method or equipment. Several studies have established reference values for MIP and MEP across different populations, including those in the Middle East [3], Brazil [4,5], Colombia [6], Indonesia [7], Korea [8], and Thailand [9]. These studies highlight the variability in MIP and MEP across ethnicities, age groups, and levels of physical activity [2]. These NRV may not be directly applicable to Singapore, which is a multi-ethnic nation with a rapidly ageing population. Moreover, the MIP and MEP of healthy Singaporean adults were last published in 1997 [10]. The periodic revision of NRV is essential to ensure its continued relevance and applicability across populations [11]. Demographic transitions, including ageing populations and the rising prevalence of non-communicable diseases, have significantly altered baseline health parameters over time [12]. Due to the lack of recent large-scale local data on MIP and MEP in Singapore, the accuracy of interpreting these values may be limited, which in turn affects clinical decision-making during therapeutic interventions.

While spirometry remains the gold standard for pulmonary function testing globally, its accessibility and feasibility in community and primary care settings can be limited in some countries [2,13]. Although Singapore is considered a developed city-state, where accessibility to spirometry is readily available through both private and public healthcare providers [14], residents will still have to incur additional healthcare costs if they wish to have their MIP and MEP measured. This underscores the need for alternative predictive tools that utilise easily obtainable, non-spirometric variables such as gender, anthropometric measurements and physical function indicators [7,8]. The development of predictive models based on these non-spirometric variables could enhance the feasibility of respiratory assessment in care settings without spirometry equipment. Several demographic and anthropometric variables, including age, gender, body mass index (BMI), and chest wall expansion, have been shown to influence MIP and MEP [7,8,9,15]. However, findings across studies remain inconsistent, possibly due to methodological differences and variations in the health status or body composition of study populations. As such, there is no clear consensus on which variables most significantly affect respiratory muscle strength [2]. Lung function naturally declines with age, even in individuals without diagnosed lung disease [16]. However, longitudinal data show that lungs and airways reach peak function in early adulthood during the first 20–25 years of life, before declining due to physiological lung ageing [17]. It remains unclear, however, whether maximal inspiratory and expiratory pressure also exhibit distinct age-related changes in young adults, similar to trends observed in pulmonary function.

To address these gaps, this study aims to: (1) establish the NRV of MIP and MEP in the healthy Singaporean population aged 21 to 80 years; (2) determine the correlations of variables that could influence MIP and MEP; and (3) develop regression equations using non-spirometry variables to estimate reference values for MIP and MEP in this population. In particular, age-related differences in respiratory muscle strength were examined across decades, including a sub-analysis within the 20s age group (21–24 vs. 25–29 years), to detect potential variation within younger adults. By providing population-specific reference standards and practical predictive tools, this study aims to enhance the accuracy and accessibility of respiratory muscle strength assessment in Singapore, thereby supporting more effective clinical decision-making and therapeutic interventions.

## 2. Materials and Methods

### 2.1. Study Design and Ethics

This cross-sectional convenience sampling study was conducted between March 2023 and December 2024 in various residential districts of Singapore. The University Institutional Review Board approved this study (Project number 2023016). Each participant provided written informed consent before data collection. Personal data was anonymised to safeguard against inadvertent disclosure and security breaches. Only de-identified data were used during data analysis.

### 2.2. Sample Size Calculations and Participant Recruitment

The sample size for the cross-sectional study was calculated to establish normative reference values for maximum inspiratory pressure (MIP) and maximum expiratory pressure (MEP) in healthy adults aged 21 to 80 years of both genders in Singapore. Based on the pilot trial of the present study of 50 participants [MIP: mean 89 cmH_2_O, standard deviation (SD) 51 cmH_2_O; MEP: mean 70 cmH_2_O, SD 26 cmH_2_O], using the formula $n=\frac{z^{2}\cdot SD^{2}}{MOE^{2}}$, with Z-score = 1.96 for a 95% confidence level (two-sided), SD = 51 cmH_2_O for MIP and 26 cmH_2_O for MEP, and a margin of error (MOE) of ±5 cmH_2_O. A sample size of 400 for MIP and 104 for MEP was required. The larger sample size for MIP (n = 400) was selected to ensure precision for both measurements, assuming approximate normality of the sampling distribution, which is supported by the large sample size and the central limit theorem. The large sample size will provide sufficient power for reliable estimates if the full-scale data exhibits significant non-normality. Given the single-point data collection design, no adjustment for dropout was necessary.

After obtaining written consent, all participants completed the Physical Activity Readiness Questionnaire for Everyone (PAR-Q) [18] and the Mini-Cog [19] as a baseline evaluation, and vital sign measurements were taken before spirometry. Healthy individuals aged 21–80 years residing across Singapore were consecutively recruited. Inclusion criteria were BMI ≤ 27.5 kg/m^2^ [20], normal pulmonary function (FEV_1_ > 80% predicted, FEV_1_/FVC ≥ 70%) [21], and non-smoking status. Individuals who were smokers, pregnant and had any unresolved respiratory and cardiovascular diseases, musculoskeletal conditions, and/or other difficulties or disabilities, such as cognitive impairment, that would compromise their adherence and participation with the MIP and MEP measurements protocol were excluded from the study. Additionally, those who had just recovered from rhinitis, sinusitis, or other nasal conditions within the 30 days preceding data collection were excluded.

### 2.3. Data Collection

This study adopted a single-visit design. Demographic variables (age and gender), medical history, including COVID-19 infection status, recovery date, and any prevailing or unresolved medical conditions were obtained. Anthropometric measures, including height (Seca 213 Portable Stadiometer) and weight (Tanita HD-382), were measured. The BMI was calculated using the standard formula [22].

#### 2.3.1. Chest Circumferences

Chest expansion was assessed at two levels, the axilla and the xiphoid process, adhering to a previously published protocol [23]. Participants stood in the anatomical position with shoulders abducted to 90 degrees and both hands placed behind their heads. Standardised instruction was provided: “Breathe in maximally and make yourself as big as possible” for maximal inspiration, while maximal expiration was measured with the command “Breathe out maximally and make yourself as small as possible” to ensure accurate measurement of the chest circumference during full expiration. Participants held their breath for up to 5 s, and their chest circumferences were recorded. Three measurements were taken at each level. Chest expansion at each level was calculated using the difference between the inspiratory and expiratory circumferences, with the highest recorded change in circumference used for data analysis.

#### 2.3.2. Waist–Hip Ratio

The waist–hip ratio was assessed by measuring the waist and hip circumferences following standardised procedures [24]. Participants were instructed to exhale normally before measurement. Waist circumference was recorded at the midpoint between the superior iliac crest and the inferior margin of the lowest palpable ribs. The hip circumference was measured horizontally around the broadest part of the buttocks. Each measurement was repeated twice to calculate the mean value, and the waist–hip ratio was subsequently determined.

#### 2.3.3. Maximum Inspiratory and Expiratory Pressure Assessment

Following the American Thoracic Society and European Respiratory Society (ATS/ERS) guidelines to minimise air leakage [1], the MIP and MEP were measured using a Pony FX MIP/MEP Spirometer (Cosmed USA, Chicago, IL, USA) with its manufacturer-supplied antibacterial filter and rubber mouthpieces. MIP and MEP measurements were recorded in centimetres of water (cmH_2_O) to the nearest 1 cmH_2_O.

Testing was conducted by trained investigators, including final-year physiotherapy students and licensed physiotherapists, all of whom had prior experience in respiratory muscle assessment through formal curriculum training and clinical exposure. Before data collection, all investigators underwent a standardisation session and conducted pilot testing (n = 50) to ensure consistency in test administration. Standardised verbal instructions and demonstrations were provided before each test, and participants completed a familiarisation trial to reduce learning effects [1].

Participants were seated upright in a stable chair with armrests, with both feet flat on the floor, in a position similar to that used for spirometry assessments. Participants performed a minimum of three and a maximum of eight acceptable manoeuvres, initiated from residual volume for MIP and total lung capacity for MEP, respectively. During each manoeuvre, participants were instructed to sustain end-inspiratory (for MIP) and end-expiratory (for MEP) holds for 1 to 2 s to sustain the static pressures [25]. A nose clip was worn [13], and participants were instructed to maintain a tight seal with their lips around the mouthpiece; no hand support around the mouth was used. A rest period of at least 1 min was enforced between each trial to minimise fatigue, and a 5 min rest break was taken between the MIP and MEP measurements [9].

The highest MIP and MEP values from acceptable manoeuvres were chosen for analysis, provided they were within 10% of the second-highest value and exhibited less than 5% variability across attempts, ensuring repeatability and reproducibility [9]. Manoeuvres were deemed acceptable if they were free from air leaks and displayed an apparent plateau on the pressure-time graph. No visual feedback was provided to participants during testing to avoid introducing variability in effort. However, participants performed a minimum of three and up to eight manoeuvres, allowing sufficient opportunity to achieve maximal respiratory pressures. Standardised verbal encouragement (“more, more, more”) was provided during each manoeuvre to maximise effort, ensuring that recorded values reflect true maximal performance.

### 2.4. Statistical Analysis

GraphPad Prism ver. 8.4.6 performed the statistical analysis, with a *p*-value of less than 0.05 considered statistically significant. Demographics, anthropometric and respiratory variables were assessed for normality using the Kolmogorov–Smirnov test. Descriptive statistics, including the median and interquartile range (IQR), were used to evaluate central tendency and data spread. The Mann–Whitney U test compared the variables between genders, with effect size determined using the rank-biserial correlation (r_rb). The r_rb, ranging from −1 to 1, measures the magnitude of the difference in ranks between groups. Values of 0 to ±0.3 indicate a small effect, ±0.3 to ±0.6 a moderate effect, and ±0.6 to ±1 a large effect. The Kruskal–Wallis and Bonferroni tests analysed non-normally distributed continuous variables across different age groups. The Spearman correlation coefficient was used to establish the correlation between variables. Values of 0 to ±0.3 indicate a weak correlation, ±0.3 to ±0.6 a moderate correlation, and ±0.6 to ±1 a strong correlation. Multivariate regression was employed to derive the reference equations for MIP and MEP based on the collected data.

## 3. Results

### 3.1. Participants

Figure 1 describes the participant recruitment process, and Table 1 presents the demographics and characteristics of the participants. Three hundred ninety-one participants (202 females, 189 males) completed the present study. The median age of all participants was 26.0 years (IQR 23.5 to 36), with an age range of 21 to 77 years. The overall median of MIP was 94 cmH_2_O (IQR 72 to 121) [female: 83 cmH_2_O (IQR 65 to 103); male 112 cmH_2_O (IQR 88 to 132), *p* < 0.001], and median of MEP was 74 cmH_2_O (IQR 61 to 88) [female: 64 cmH_2_O (IQR 56 to 75); male 85 cmH_2_O (IQR 72 to 106), *p* < 0.001].

### 3.2. MIP and MEP Values

Figure 2 and Figure 3 and Table 2 illustrate the NRV of MIP and MEP in healthy Singaporean adult participants recruited for the present study. The analyses aimed to explore potential age variations in MIP and MEP to sub-stratify participants in their 20s into two age groups (21–24 and 25–29 years). This is based on prior research on spirometry and respiratory function, which suggests that age-related changes within this decade may yield insightful analyses [16,17,26]. The mean difference in MIP between the 21–24 and 25–29 age groups was −14.96 cmH_2_O (standard error = 14.61). Applying the Bonferroni correction for multiple comparisons, no statistically significant difference was observed (t = −1.02, adjusted *p* = 1.000). Similarly, the mean difference in MEP between the 21–24 and 25–29 age groups was −15.92 cmH_2_O (standard error = 14.60), no statistically significant difference was found (t = −1.09, adjusted *p* = 1.000) after applying Bonferroni correction. Interestingly, the median MIP from the 30–39 female subgroup [73.0 cmH_2_O (IQR 55.0–88.0)] was unexpectedly lower than that for females aged 25–29 years (89.5 cmH_2_O) and 40–49 years (93.0 cmH_2_O), indicating a notable dip that warrants further investigation.

### 3.3. Correlation with MIP and MEP

Table 3 presents the Spearman correlation coefficient between the measured variables with MIP and MEP. The number of days since the last COVID-19 infection was the only variable that did not correlate with MIP or MEP in the examined cohort.

### 3.4. Prediction of MIP and MEP

Table 4 reports the multiple linear regression model and equation for predicting MIP and MEP using readily available information, including age, gender, height and weight. The MIP prediction equation accounted for 27.5% of the variance in maximal inspiratory pressure, while the MEP prediction equation accounted for 32.1% of the variance in maximal expiratory pressure.

## 4. Discussion

This study updated the NRV for MIP and MEP in a healthy Singaporean adult population aged 21 to 80 years. These values serve as an essential revision to the only existing local report published in 1997 [10], reflecting the substantial demographic shifts and changes in population health profiles over the past two decades [12]. Compared to the earlier report by Johan et al. (1997) [10], our findings reveal notable differences in both MIP and MEP values. For MIP, current values were substantially higher in both females (83.0 cmH_2_O vs. 53.6 cmH_2_O) and males (112 cmH_2_O vs. 88.7 cmH_2_O), indicating an overall improvement in inspiratory muscle strength over the past two decades. In contrast, MEP values show a divergent pattern: while males in the present study recorded lower MEP values than previously reported (85 cmH_2_O vs. 113.4 cmH_2_O), females showed a marginal difference (Females: 64.0 cmH_2_O vs. 68.3 cmH_2_O). These findings suggest that while inspiratory muscle strength appears to have increased in the current population, expiratory strength may not have followed the same trend, particularly among males. This shift may reflect changes in physical activity patterns, lifestyle habits, or other unmeasured health-related factors that have evolved across generations. Additionally, it is important to acknowledge that the age profile of the study sample was not directly comparable. In Johan’s earlier study, the male participants had mean ages of 40.8 (± 13.4), 37.3 (± 11.5), and 39.7 (± 11.5) years across different ethnicity subgroups [10], while in the current study, the median age of male participants was notably younger at 26.0 (IQR 24.0 to 35.0) (Table 1). Similarly, for females, the median age in our cohort was 25.5 years (IQR 23.0 to 36.8), compared to mean ages of 38.9 (± 11.8), 33.4 (± 9.6), and 35.1 (± 11.8) in Johan’s report [10]. These age differences may partially account for the observed variation in respiratory pressures.

Interestingly, our findings demonstrated consistently higher MIP values compared to MEP across all age groups and in both genders. While previous studies, such as that by Simões et al. (2010), have documented instances where MIP exceeded MEP in healthy individuals, specifically those aged 50 years and above [5], these are typically based on age or lifestyles, or presented as individual observations rather than population-level normative references. To the best of our knowledge, the present study is among the first to report this trend consistently across a large cohort and stratified age groups. No other published normative reference studies, particularly those involving healthy adult populations, have consistently reported higher MIP than MEP values across age and gender. In view of the atypical finding, where MIP values exceeded MEP across all subgroups, the investigators, cognisant of established physiological expectations, conducted a meticulous inspection of the raw data to verify the integrity and accuracy of the results. Upon raw data inspection, it was found that 82 out of 391 participants recorded MEP values greater than MIP, comprising 47 females and 35 males. These individuals were distributed across the age groups as follows: 20s (n = 42, 16%), 30s (n = 16, 27%), 40s (n = 8, 24%), 50s (n = 9, 30%), and 60s (n = 7, 28%). Additionally, eight participants (five females, three males) recorded equal MIP and MEP values; all but one were in their 20s, with the remaining participant in his 30s. The descriptive statistics in the present study indicate an overall trend of higher MIP compared with MEP. The inspection of the raw data revealed variability at the individual level, with some participants demonstrating MEP values exceeding MIP, others with equal values, and some with MEP lower than MIP, despite this group representing a majority in the current sample. Thus, another set of graphs were plotted to compare those aged 21 to 49 years with those older than 50 years. (Appendix A). The trends remained consistent, with the MEP being lower than the MIP. The medians of the MIP were [Male (age 21–49) 113.5 cmH2O (IQR 90.0 to 135.3); (age ≥ 50) 87 cmH2O (IQR 72.3 to 107.0)]; while the medians of the MEP were [Male (age 21–49) 87.0 cmH2O (IQR 72.3 to 107.0); Male (age ≥ 50) 80.0 cmH2O (IQR 70.0 to 102.0)]. Similarly for the female participants, the medians of the MIP were [Female (age 21–49) 85.0 cmH2O (IQR 67.0 to 105.3); Female (age ≥ 50) 65.5 cmH2O (IQR 52.3 to 85.0)], and the medians of the MEP were [Female (age 21–49) 65.5 cmH2O (IQR 56.0 to 78.0); Female (age ≥ 50) 57.5 cmH2O (IQR 52.3 to 64.5)]. The overall trend may therefore reflect the influence of statistical aggregation, as most participants recorded higher MIP values. Given the majority representation with the sampled participants in this study, different physiological and anatomical factors, such as lung volume at testing, muscle architecture, physical conditioning, and chest wall mechanics, are known to influence MIP and MEP [27], these cannot be uniformly applied to explain the present study’s findings. However, the authors postulate that a possible explanation could be that the diaphragm, the primary muscle of inspiration, is stronger and more efficiently recruited compared to the expiratory muscles in this predominantly younger sample, who are possibly more physically active. Aligning with past findings that younger age is significantly associated with higher MIP values in healthy adults [15]. Despite this possible explanation, given the characteristics of our participant cohort, no single factor or combination can sufficiently account for the observed pattern of higher MIP than MEP, suggesting that additional or population-specific influences may be at play. These influences may include lifestyle factors, higher physical activity levels, or population-specific anatomical characteristics that favour inspiratory over expiratory muscle performance.

### 4.1. Gender and Age Differences for MIP and MEP

Despite the unique observation in the present study, the gender and age-related patterns align well with established literature [10,15]. Males demonstrated higher values than females for both MIP [Males: 112 cmH_2_O (88 to 132) vs. Females: 83 cmH_2_O (65 to 103), *p* < 0.001, r_rb = 0.40] and MEP [Males: 85 cmH_2_O (72 to 106) vs. Females: 64 cmH_2_O (56 to 75), *p* < 0.001, r_rb = 0.54]. Additionally, a clear age-related decline in respiratory muscle strength was observed, with MIP measures decreasing progressively across older age groups (Table 2). This trend is comparable to those reported in other Asian populations [8,9], but slightly lower than values reported in Caucasian cohorts [2,4], underscoring the importance of population-specific reference standards. Interestingly, both MIP and MEP generally peaked in the 40–49 age range in the current dataset. This finding aligns with data from a Korean cohort [8] but contrasts with observations from Caucasian populations, where peak values tend to occur earlier [2,4]. The inconsistent pattern of increase and decrease in MIP and MEP up to the fourth decade remains unclear. It is possible that cohort-specific factors such as lifestyle, health status, or physical activity levels may influence these trends [2,8]. An unexpected reduction in MIP was observed among females aged 30–39 years (Table 2). While the underlying cause is unclear, this trend may reflect transitional physiological or specific lifestyle differences among the sampled participants, such as alterations in physical activity levels, familial or occupational demands during this life stage. However, given the relatively small subgroup size (n = 29), this could be due to a sampling issue, and the findings should be interpreted with caution. Further research is needed to determine whether this pattern is consistent across larger and more diverse cohorts.

This study specifically aimed to investigate whether differences in respiratory muscle strength exist within early adulthood by comparing participants aged 21–24 and 25–29 years. While gender, demographic, and anthropometric factors were considered, the findings revealed no statistically significant differences in MIP and MEP between these two subgroups. This suggests that respiratory muscle strength remains relatively stable throughout the third decade of life. These findings align with previous spirometry-based studies, which have similarly reported minimal physiological variation in pulmonary function during early adulthood [16,17,26].

### 4.2. Correlations of MIP and MEP

Spearman correlation analyses revealed that gender, height, weight, BMI, waist–hip ratio and pulmonary function variables (FVC, FEV_1_, PEF) were significantly associated with both MIP and MEP. These findings are consistent with prior studies that identified anthropometric and spirometric variables as key determinants of respiratory muscle strength [3,9,15].

Chest circumferences at the axilla and xiphoid levels correlated significantly with MIP, consistent with previous studies, which demonstrated that thoracic expansion measurements are predictive of MIP values [7,23]. The lack of correlation between chest circumference and MEP may reflect the complex interplay between muscle strength, lung volume, and thoracic mechanics. While MEP is measured at total lung capacity (TLC), where expiratory muscles are theoretically at their optimal mechanical advantage, chest circumference alone may not adequately represent the functional capacity or recruitment efficiency of the abdominal and internal intercostal muscles. Additionally, MEP generation is influenced by factors such as neuromuscular coordination, abdominal wall compliance, and expiratory effort technique, which may not correlate directly with static anthropometric measures like chest circumference [1].

Severin and colleagues [28] concluded that patients with severe COVID-19 infection may experience respiratory muscle damage, contributing to persistent dyspnoea regardless of the time elapsed since infection. In contrast, the healthy cohort in this study did not report persistent dyspnoea following COVID-19, suggesting an absence of respiratory muscle damage. This distinction may help explain why the number of days since the last COVID-19 infection did not correlate with MIP or MEP in this group.

### 4.3. Regression Equations and Clinical Utility

The regression models developed in this study accounted for 27.5% and 32.1% of the variance in MIP and MEP, respectively. While the regression equations provide practical tools for estimating respiratory muscle strength using non-spirometry variables, the relatively modest R^2^ values indicate that other unmeasured factors, such as physical activity levels, muscle composition, or socioeconomic and nutritional status, may influence MIP and MEP values [8,9]. Nonetheless, the regression equations offer a valuable alternative for settings where spirometry is unavailable or impractical [2,13]. Possible clinical applications include approximating normal respiratory muscle strength, providing benchmarks to indicate suboptimal respiratory pressure and guiding prescriptions for therapeutic interventions, such as inspiratory muscle training.

### 4.4. Limitations

Several limitations should be acknowledged. First, although the sample size was sufficient and recruited from a community-based population, 91.5% of participants identified as Chinese, a proportion notably higher than the national average (approximately 74%). This likely reflects the demographic composition of the recruitment sites rather than the broader population distribution. Nonetheless, the predominance of Chinese participants limits the generalisability of the findings to other ethnic groups. This is particularly important given prior local research reported that Chinese individuals in Singapore tend to have higher MIP and MEP values than other ethnic groups [10]. As such, population-specific differences must be considered, and future studies should aim to include more ethnically diverse samples to enhance representativeness. Second, the median age of all participants was 26.0 years despite an age range of 21 to 77 years. Older adults (>60 years) were relatively underrepresented, which may affect the robustness of the NRV in this subgroup. As age is a key determinant of respiratory muscle strength, the generalisability of the findings and predictive models to older adults should be interpreted with caution. Thirdly, while the regression models identified several significant predictors, the relatively modest R^2^ values (27.5% for MIP and 32.1% for MEP) indicate that a substantial proportion of variance remains unexplained. This likely reflects the study’s focus on examining only demographic and anthropometric variables, in line with its defined objectives. Other potentially relevant factors, such as physical activity levels, respiratory muscle composition, neuromuscular coordination, and socioeconomic or nutritional status, were not assessed and may also influence MIP and MEP. Future studies should consider incorporating these variables to improve model performance and predictive accuracy.

### 4.5. Future Directions

Future research should aim to recruit samples that more closely reflect Singapore’s national ethnic composition to enhance the representativeness of the normative values established. Greater inclusion of older adults is also recommended. Furthermore, integrating additional variables, such as habitual physical activity, respiratory muscle composition, nutritional status, and socioeconomic background, may help explain the variance not captured in the current predictive models. Longitudinal studies may also be explored to track changes in MIP and MEP over time and to refine predictive models for clinical use. In addition, the observation of higher median MIP compared to MEP across gender and age groups, contrary to conventional expectations, warrants further exploration. While the measured demographics or anthropometric factors did not fully explain this finding, it highlights the need for more detailed physiological investigations, including respiratory muscle recruitment patterns, lung volume calibration, or potential methodological influences.

## 5. Conclusions

This study updated the normative reference values (NRV) for maximal inspiratory pressure (MIP) and maximal expiratory pressure (MEP) in a healthy adult Singaporean population. Demographic and anthropometric variables, including gender, height, weight, BMI, and waist-to-hip ratio, were significantly correlated with both MIP and MEP. The development of regression equations using non-spirometry variables offers practical alternatives for a more accessible approximation of respiratory muscle strength in settings where spirometry is not readily available, thereby guiding prescriptions for therapeutic interventions such as inspiratory muscle training. While these findings offer valuable insights, future studies should validate the predictive models in older and more ethnically diverse populations and further explore atypical trends, such as the unexpectedly higher MIP over MEP.

## Figures and Tables

**Figure 1 muscles-04-00047-f001:**
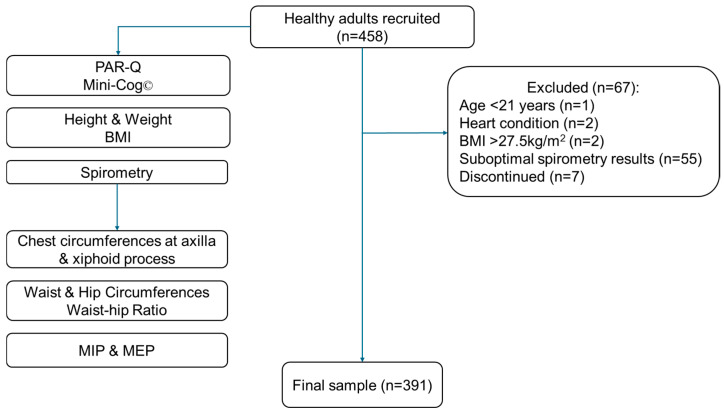
Participant recruitment process (Note: PAR-Q: Physical Activity Readiness Questionnaire for Everyone; BMI: body mass index; MIP: maximal inspiratory pressure; MEP: maximal expiratory pressure).

**Figure 2 muscles-04-00047-f002:**
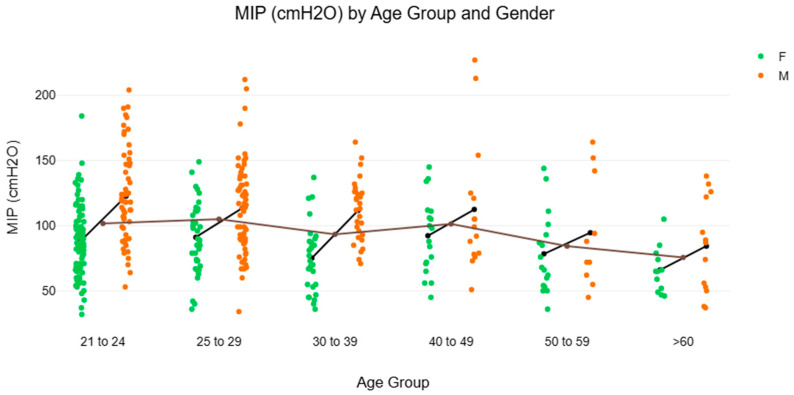
Age Group vs. Maximal Inspiratory Pressure: Median and Maximum-Minium Range Comparison by Gender (Note: F: Female; M: male; MIP: maximal inspiratory pressure; cmH_2_O: centimetre of water. Participant distribution by age group and gender: 20–24 years F: n = 88, M: n = 59; 25–29 years F: n = 38, M: n = 63; 30–39 years F: n = 29, M: n = 29; 40–49 years F: n = 18, M: n = 15; 50–59 years F: n = 17, M: n = 10; >60 years F: n = 12, M: n = 13).

**Figure 3 muscles-04-00047-f003:**
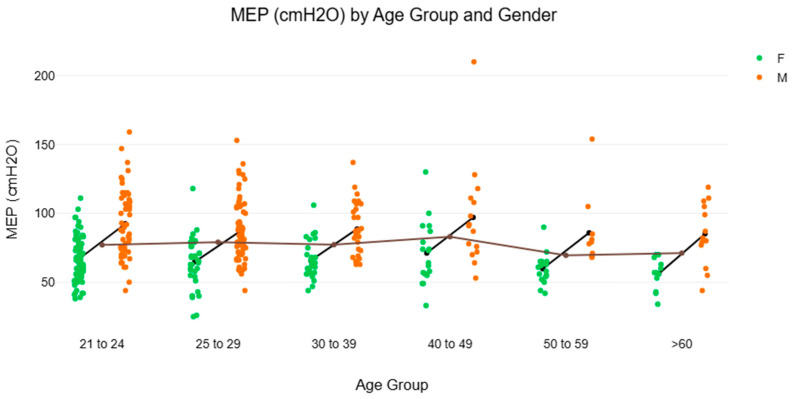
Age Group vs. Maximal Expiratory Pressure: Median and Maximum-Minium Range Comparison by Gender (Note: F: Female; M: male; MEP: maximal expiratory pressure; cmH_2_O: centimetre of water. Participant distribution by age group and gender: 20–24 years F: n = 88, M: n = 59; 25–29 years F: n = 38, M: n = 63; 30–39 years F: n = 29, M: n = 29; 40–49 years F: n = 18, M: n = 15; 50–59 years F: n = 17, M: n = 10; >60 years F: n = 12, M: n = 13).

**Table 1 muscles-04-00047-t001:** Characteristics of participants and measured variables (by gender).

Variables	Total (n = 391)	Female (n = 202)	Male (n = 189)	*p*-Value	Rank- Biserial Correlation
Age, years	26.0 (23.5 to 36.0)	25.5 (23.0 to 36.8)	26.0 (24.0 to 35.0)	0.052	0.10
Height, cm	165 (160 to 172)	160 (156 to 165)	172 (167 to 176)	<0.001	0.70
Weight, kg	60.3 (53.6 to 70.1)	54.5 (50.4 to 59.8)	69.3 (61.7 to 75.8)	<0.001	0.66
BMI, kg/m^2^	22.0 (20.3 to 24.4)	21.2 (19.8 to 22.8)	23.4 (21.5 to 25.2)	<0.001	0.36
Race, n (%)					
Chinese	358 (91.5)	192 (49.1)	166 (42.4)		
Malay	18 (4.6)	7 (1.8)	11 (2.8)		
Indian	7 (1.8)	1 (0.3)	6 (1.5)		
Eurasian	1 (0.3)	1 (0.3)	0 (0.0)		
Others:	7 (1.8)	1 (0.3)	6 (1.5)		
COVID-19, occurrences	1 (1 to 1)	1 (1 to 1.8)	1 (1 to 1)	0.945	0.00
Days since the last COVID-19 infection	437 (219 to 595)	448 (203 to 596)	428 (236 to 594)	0.887	0.01
Chest circumference (axilla), cm	4.4 (3.3 to 5.5)	4.4 (3.2 to 5.4)	4.5 (3.3 to 5.6)	0.574	0.03
Chest circumference (xiphoid), cm	5.6 (4.1 to 7.0)	5.4 (3.8 to 6.7)	5.9 (4.4 to 7.5)	0.004	0.14
Waist–hip ratio	0.8 (0.8 to 0.9)	0.8 (0.7 to 0.8)	0.8 (0.8 to 0.9)	<0.001	0.39
FVC, litres	3.6 (3.1 to 4.3)	3.2 (2.9 to 3.5)	4.3 (3.8 to 4.8)	<0.001	0.71
FEV_1_, litres	3.2 (2.7 to 3.7)	2.8 (2.5 to 3.1)	3.7 (3.4 to 4.1)	<0.001	0.69
FEV_1_/FVC%	88 (84 to 91)	89 (85 to 91.8)	86 (82 to 91)	0.692	0.02
PEF	7.3 (6.2 to 9.0)	6.3 (5.5 to 7.1)	9.0 (7.9 to 10.0)	<0.001	0.72
PEF_25–75%_	3.6 (3.0 to 4.3)	3.3 (2.9 to 3.9)	4.2 (3.3 to 5.0)	<0.001	0.42
MIP, cmH_2_O	94 (72 to 121)	83 (65 to 103)	112 (88 to 132)	<0.001	0.40
MEP, cmH_2_O	74 (61 to 88)	64 (56 to 75)	85 (72 to 106)	<0.001	0.54

Values are expressed as median and interquartile range (IQR), *p* < 0.05 represents a significant value; n: sample size; cm: centimetres, kg: kilograms, BMI: body mass index; FVC: forced vital capacity; FEV_1_: Forced expiratory volume at the end of first second; FEV_1_/FVC%: the percentage of the FVC expired in one second; PEF: peak expiratory flow; PEF_25–75%_: peak expiratory flow at 25% and 75% of the pulmonary volume; MIP: maximum inspiratory pressure; MEP: maximum expiratory pressure; cmH_2_O: centimetre of water.

**Table 2 muscles-04-00047-t002:** Maximum inspiratory pressure and maximum expiratory pressure stratified by age and gender.

Age Group	MIP, cmH_2_O	*p*-Value	MEP, cmH_2_O	*p*-Value
20 to 24				
Female (n = 88)	86.0 (66.8 to 103.3)	<0.001	66.0 (56.0 to 77.0)	<0.001
Male (n = 59)	118.0 (98.5 to 147.0)		88.0 (73.5 to 109.5)	
25 to 29				
Female (n = 38)	89.5 (73.0 to 111.0)	<0.001	67.0 (55.8 to 78.0)	<0.001
Male (n = 63)	114.0 (89.0 to 134.0)		83.0 (71.5 to 102.5)	
30 to 39				
Female (n = 29)	73.0 (55.0 to 88.0)	<0.001	63.0 (57.0 to 73.0)	<0.001
Male (n = 29)	113.0 (93.0 to 126.0)		87.0 (73.0 to 103.0)	
40 to 49				
Female (n = 18)	93.0 (71.3 to 109.8)	<0.001	68.5 (56.3 to 85.0)	<0.001
Male (n = 15)	99.0 (78.5 to 123.0)		91.0 (74.0 to 109.5)	
50 to 59				
Female (n = 17)	68.0 (54.0 to 93.0)	<0.001	61.0 (54.0 to 65.0)	<0.001
Male (n = 10)	80.0 (64.5 to 130.0)		78.5 (69.5 to 84.0)	
>60				
Female (n = 12)	65.0 (51.3 to 75.3)	<0.001	57.0 (50.5 to 64.3)	<0.001
Male (n = 13)	87.0 (53.0 to 122.0)		82.0 (77.0 to 105.0)	

**Table 3 muscles-04-00047-t003:** Spearman correlation coefficient (r) for maximum inspiratory and expiratory pressures with measured variables.

Variables	Maximum Inspiratory Pressure			Maximum Expiratory Pressure		
	r	95% CI	*p*-Value	r	95% CI	*p*-Value
Gender	−0.399	−0.482 to −0.310	<0.001	−0.537	−0.606 to −0.460	<0.001
Age, years	−0.149	−0.247 to −0.0476	0.003	−0.022	−0.124 to 0.080	0.660
Height, cm	0.307	0.212 to 0.397	<0.001	0.368	0.276 to 0.453	<0.001
Weight, kg	0.412	0.323 to 0.493	<0.001	0.455	0.370 to 0.532	<0.001
BMI, kg/m^2^	0.356	0.264 to 0.442	<0.001	0.362	0.270 to 0.448	<0.001
Days since the last COVID-19 infection	−0.0632	−0.175 to 0.050	0.260	−0.031	−0.143 to 0.082	0.580
Chest circumference (axilla), cm	0.187	0.086 to 0.283	<0.001	0.078	−0.025 to 0.178	0.130
Chest circumference (xiphoid), cm	0.209	0.109 to 0.305	<0.001	0.152	0.050 to 0.250	0.003
Waist–hip ratio	0.202	0.102 to 0.298	<0.001	0.292	0.195 to 0.382	<0.001
FVC, litres	0.408	0.319 to 0.489	<0.001	0.420	0.332 to 0.500	<0.001
FEV_1_, litres	0.404	0.315 to 0.486	<0.001	0.407	0.318 to 0.488	<0.001
PEF	0.509	0.429 to 0.581	<0.001	0.544	0.468 to 0.612	<0.001
PEF_25–75%_	0.229	0.072 to 0.374	0.004	0.270	0.116 to 0.411	<0.001

*p* < 0.05 represents a significant value; cm: centimetres, kg: kilograms, BMI: body mass index; FVC: forced vital capacity; FEV_1_: Forced expiratory volume at the end of first second; PEF: peak expiratory flow; PEF_25–75%_: peak expiratory flow at 25% and 75% of the pulmonary volume.

**Table 4 muscles-04-00047-t004:** Multiple linear regression model and equation for predicting maximum inspiratory and expiratory pressure with non-spirometry variables.

MIP = 257.1 − 23.4 (Female = 1; Male = 0) − 0.72 (Age, Year) − 1.27 (Height, cm) + 1.41 (Weight, kg)				
R^2^ = 27.5%	Coefficient	Standard Error	*p*-Value	95% CI
Constant	257.1	46.51	<0.001	165.6 to 348.5
Gender	−23.4	4.33	<0.001	−31.9 to −14.9
Age, year	−0.72	0.13	<0.001	−0.97 to −0.47
Height, cm	−1.27	0.31	<0.001	−1.88 to −0.67
Weight, kg	1.41	0.23	<0.001	0.97 to 1.86
**MEP = 146.8 − 20.1 (female = 1; male = 0) − 0.20(Age, year) − 0.63(Height, cm) + 0.83(Weight, kg)**				
**R^2^ = 32.1**	**Coefficient**	**Standard error**	** *p* ** **-value**	**95% CI**
Constant	146.8	30.35	<0.001	87.2 to 206.5
Gender	−20.1	2.82	<0.001	−25.7 to −14.6
Age, year	−0.20	0.08	=0.010	−0.36 to −0.04
Height, cm	−0.63	0.20	=0.002	−1.02 to −0.23
Weight, kg	0.83	0.15	<0.001	0.54 to 1.12

*p* < 0.05 represents a significant value; cm: centimetres, kg: kilograms; 95% CI: 95% confidence interval.

## Data Availability

The data supporting the findings of this study are openly available in figshare at https://figshare.com/s/29d80b671ea903177fe0 (accessed on 29 June 2025).

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
