# Peer review of "Maximal Respiratory Muscle Strength in Singaporean Adults: Normative Reference Values and Predictive Models from a Cross-Sectional Study"

_muscles, 2025, doi:10.3390/muscles4040047_

Round 1

Reviewer 1 Report

Comments and Suggestions for Authors

This manuscript presents a well-conducted cross-sectional study that updates normative reference values (NRVs) for maximal inspiratory (MIP) and expiratory (MEP) pressures in healthy Singaporean adults. The authors also derive predictive equations using non-spirometry variables, which has practical value in clinical and field settings.

The manuscript is timely and well-written. No major methodological concerns were identified, but several minor clarifications will improve the quality and generalizability of the work:

  1. Ethnic Representation: While 91.5% of the sample were Chinese, which is above the national average (~74%), this reflects local demographics in urban academic settings. Still, the limitation in generalizability to other ethnic groups should be acknowledged more explicitly.

  2. Regression Models: The R² values (27.5% for MIP, 32.1% for MEP) suggest other unmeasured variables (e.g., activity level, muscle composition) influence results. Acknowledge these as future predictors.

  3. Older Adult Representation: The median age was 26, and participants over 60 were underrepresented. State whether models should be applied cautiously in this group.

  4. MIP Dip in Females 30–39: Provide a brief interpretation of the unexpected dip in MIP among females in this age group.

  5. Terminology: Consider briefly defining “NRV” in the Introduction for clarity.

  6. Figures: Add sample size labels per age group in Figures 2 and 3 to improve readability.

Author Response

Dear Reviewer,

Thank you for the opportunity to revise our manuscript entitled “Maximal Respiratory Muscle Strength in Singaporean Adults: Normative Reference Values and Predictive Models from a Cross-Sectional Study.” We appreciate the thoughtful feedback provided by you, which has helped us to improve the clarity, rigour, and overall quality of our work. In this letter, we detail our responses to each comment and outline the corresponding revisions made to the manuscript. We trust that these changes address the concerns raised and enhance the manuscript’s contribution to the field.

This manuscript presents a well-conducted cross-sectional study that updates normative reference values (NRVs) for maximal inspiratory (MIP) and expiratory (MEP) pressures in healthy Singaporean adults. The authors also derive predictive equations using non-spirometry variables, which has practical value in clinical and field settings.

The manuscript is timely and well-written. No major methodological concerns were identified, but several minor clarifications will improve the quality and generalizability of the work:

Response: We thank you for the encouragement.

  1. Ethnic Representation: While 91.5% of the sample were Chinese, which is above the national average (~74%), this reflects local demographics in urban academic settings. Still, the limitation in generalizability to other ethnic groups should be acknowledged more explicitly.

Response: Thank you for the suggestion. We have included the point in Limitations.

  1. Regression Models: The R² values (27.5% for MIP, 32.1% for MEP) suggest other unmeasured variables (e.g., activity level, muscle composition) influence results. Acknowledge these as future predictors.

Response: Thanks once again. The additional point is now mentioned in the Limitations section.

  1. Older Adult Representation: The median age was 26, and participants over 60 were underrepresented. State whether models should be applied cautiously in this group.

Response: Thanks again. We have included the additional point for cautious interpretations.

  1. MIP Dip in Females 30–39: Provide a brief interpretation of the unexpected dip in MIP among females in this age group.

Response: Thanks for the insightful comment. We have included a small section to discuss this finding in the Discussion.

  1. Terminology: Consider briefly defining “NRV” in the Introduction for clarity.

Response: The additional definition is included in the Introduction.

  1. Figures: Add sample size labels per age group in Figures 2 and 3 to improve readability.

Response: It is our intention not to include the sample size label, hoping to keep the figures “less busy” looking. Keeping to our original intention, the sample size information is added under the Figure description now for Figures 2 and 3, instead.  

Reviewer 2 Report

Comments and Suggestions for Authors

Dear authors

I read your manuscript with great interest. Below you will find some doubts and suggestions that I hope can contribute to the final version.

- line 35-36: could you explain that? Measurement technique is standardized in guidelines. And, how population specific reference value will control the technical variability?
- line 51: Is that the case of Singapore? I think that a national study should be referenced.
- line 59: the gender is well established, and reflected in several reference values.
- Line 63: the study you cited did not exclude pathologies. So, I think that you should rethink this statement or find an appropriate reference.
- line 63-67: you studied healthy people, so taking into account what I said earlier, I think this sentence should be rethought.
- line 90-91: PImax had a larger variability? I think that subjects normally have more difficulty performing PImax that PEmax.
- line 102-105: this sentence reinforces what I pointed out previously, you are studying healthy subjects without major changes in weight. Therefore, lines: 63-67 should be rethought.
- line 111-113: this phrase seems incomplete.
- line 138: Since a spirometer doesn't measure MIP and MEP, you should describe the device that was used.
- line 142: it would be important that you specify this. What was the background? How long was the training program? Training is one thing but expertise (since you aim to develop reference values) is a completely different matter.
- line 145: what kind of mouthpiece was used? Because is not irrelevant. There was a leak as recomended (guidelines ATS/ERS, 2002)? Participants put their hands around their lips, for a good sealing?
- line 159: I’m sorry but I can’t agree. Since you want maximal pressures, and aim to develop reference values, you should do everything that you can to achieve them.
- figure 1: why "lung" spirometry?
- line 195: this was not stated before
- line 198: related to reference 17, I already comment on this. I recommend to check it.
- line 245-246: could you discuss the fact that you have lower MEPs than MIPs?
- line 260: taking into account what I said before, this result was expected, don't you agree?
- line 270-271: I agree, but MEP is measured at TLC, so how do your explain you results?

Kind regards

Author Response

 Dear Reviewer,

We are grateful for the opportunity to revise our manuscript titled “Maximal Respiratory Muscle Strength in Singaporean Adults: Normative Reference Values and Predictive Models from a Cross-Sectional Study.” We sincerely thank you for the constructive and insightful comments, which have guided us in refining our work. In the sections below, we provide a point-by-point response to each comment and describe the corresponding revisions made to the manuscript. We hope that these amendments satisfactorily address the concerns raised and strengthen the manuscript’s scientific contribution.

I read your manuscript with great interest. Below you will find some doubts and suggestions that I hope can contribute to the final version.

- line 35-36: could you explain that? Measurement technique is standardized in guidelines. And, how population specific reference value will control the technical variability?

Response: Thanks for the comments. Please see the revised section on standardised technique in the guideline highlighted in yellow.

- line 51: Is that the case of Singapore? I think that a national study should be referenced.

Response: We have revised this section with a local reference, highlighted in yellow.

- line 59: the gender is well established, and reflected in several reference values.

Response: Thanks very much for the useful prompt. We have included gender as an established variable.

- Line 63: the study you cited did not exclude pathologies. So, I think that you should rethink this statement or find an appropriate reference.

Response: Thanks for the insightful comment. We have replaced this with an appropriate reference, which includes healthy individuals.

- line 63-67: you studied healthy people, so taking into account what I said earlier, I think this sentence should be rethought.

Response: Thanks once again. We have revised the sentences with these highlighted in yellow.

- line 90-91: PImax had a larger variability? I think that subjects normally have more difficulty performing PImax that PEmax.

Response: Indeed, these values are derived from our initial pilot trial and the subsequent full-scale study's sample size. In additions, our results seem to record a consistently higher MIP values than the MEP. We are including additional discussion on this observation in the Discussion section.

- line 102-105: this sentence reinforces what I pointed out previously, you are studying healthy subjects without major changes in weight. Therefore, lines: 63-67 should be rethought.

Response: Thanks once again, we have revised the necessary.

- line 111-113: this phrase seems incomplete.

Response: Thanks for the highlight. We have revised the writing style of the inclusion criteria.

- line 138: Since a spirometer doesn't measure MIP and MEP, you should describe the device that was used.

Response: Thanks for highlighting the potential confusion for the future readers. However, we would like to reiterate that we used the Pony FX MIP MEP Spirometer (Cosmed USA, Chicago, IL, USA), and this spirometer is equipped with the MIP/MEP function. https://www.mesr.com.au/pony-fx-spirometer-page

- line 142: it would be important that you specify this. What was the background? How long was the training program? Training is one thing but expertise (since you aim to develop reference values) is a completely different matter.

Response: Thank you for the highlight. We provided additional details on the backgrounds of the investigators and the training/familiarisation/standardisation we put in place during the investigation.

- line 145: what kind of mouthpiece was used? Because is not irrelevant. There was a leak as recomended (guidelines ATS/ERS, 2002)? Participants put their hands around their lips, for a good sealing?

Response: We thanks you for this important clarification. As stated, we used the Pony FX spirometer (Cosmed USA, Chicago, IL, USA) with its manufacturer-supplied antibacterial filter and rubber mouthpieces, which are designed to meet ATS/ERS recommendations and minimise air leakage. All participants wore a nose clip and were instructed to maintain a tight seal with their lips; no hand sealing was required. We have added this information to the revised manuscript to improve clarity.

- line 159: I’m sorry but I can’t agree. Since you want maximal pressures, and aim to develop reference values, you should do everything that you can to achieve them.

Response: We thank you for the comment. While visual feedback can be used in some protocols, we chose not to provide it to minimise variability in participant effort due to external cues. Importantly, each participant completed between three and eight manoeuvres, with standardised verbal encouragement, which we believe provided adequate opportunity to reach maximal respiratory pressures. Therefore, we are confident that the values reported accurately represent maximal efforts.

- figure 1: why "lung" spirometry?

Response: We have revised the choice of words.

- line 195: this was not stated before

Response: We thank you for highlighting this oversight. In response, we have revised the final paragraph of the Introduction to include a clear statement of our intention to explore age-related differences in MIP and MEP, including a sub-analysis within the 20s age group (21–24 vs. 25–29 years). This sub-stratification was conducted to better understand potential variations in respiratory muscle strength within the younger adult population. The revised text now aligns the stated objectives with the analyses presented in the Results section.

- line 198: related to reference 17, I already comment on this. I recommend to check it.

Response: We appreciate the comment. Reference 17 (Karmaus et al., 2019) suggests that there may be differences in lung function trajectories within early adulthood. Based on this, we aimed to explore potential differences in respiratory muscle strength between the 21–24 and 25–29 age groups. To support this analysis, we deliberately recruited a larger number of participants within the 20s age range. Our results ultimately showed no statistically significant differences in either MIP or MEP between the two subgroups. We have clarified this rationale in the Introduction and addressed the findings appropriately in the Discussion.

- line 245-246: could you discuss the fact that you have lower MEPs than MIPs?
- line 260: taking into account what I said before, this result was expected, don't you agree?
- line 270-271: I agree, but MEP is measured at TLC, so how do your explain you results?

Responses (line 245-246) (line 260) (line 270-271): We appreciate insightful prompts to enhance our discussion. We have rewritten a large part of the discussion, hoping to address these three points more congruently. We hope our revision has addressed the issues we lacked in-depth discussion originally.

Kind regards
